# Therapeutic Potential of *Bacteroides fragilis* SNBF-1 as a Next-Generation Probiotic: In Vitro Efficacy in Lipid and Carbohydrate Metabolism and Antioxidant Activity

**DOI:** 10.3390/foods13050735

**Published:** 2024-02-28

**Authors:** Weihe Cang, Xuan Li, Jiayi Tang, Ying Wang, Delun Mu, Chunting Wu, Haisu Shi, Lin Shi, Junrui Wu, Rina Wu

**Affiliations:** 1College of Food Science, Shenyang Agricultural University, Shenyang 110866, Chinadelunmu@163.com (D.M.);; 2Liaoning Engineering Research Center of Food Fermentation Technology, Shenyang 110866, China; 3Shenyang Key Laboratory of Microbial Fermentation Technology Innovation, Shenyang 110866, China

**Keywords:** next-generation probiotics, *B. fragilis*, toxicity, probiotic function

## Abstract

This study explores the potential of aerotolerant Bacteroides fragilis (*B. fragilis*) strains as next-generation probiotics (NGPs), focusing on their adaptability in the gastrointestinal environment, safety profile, and probiotic functions. From 23 healthy infant fecal samples, we successfully isolated 56 beneficial *B. fragilis* strains. Notably, the SNBF-1 strain demonstrated superior cholesterol removal efficiency in HepG2 cells, outshining all other strains by achieving a remarkable reduction in cholesterol by 55.38 ± 2.26%. Comprehensive genotype and phenotype analyses were conducted, including sugar utilization and antibiotic sensitivity tests, leading to the development of an optimized growth medium for SNBF-1. SNBF-1 also demonstrated robust and consistent antioxidant activity, particularly in cell-free extracts, as evidenced by an average oxygen radical absorbance capacity value of 1.061 and a 2,2-diphenyl-1-picrylhydrazyl scavenging ability of 94.53 ± 7.31%. The regulation of carbohydrate metabolism by SNBF-1 was assessed in the insulin-resistant HepG2 cell line. In enzyme inhibition assays, SNBF-1 showed significant α-amylase and α-glucosidase inhibition, with rates of 87.04 ± 2.03% and 37.82 ± 1.36%, respectively. Furthermore, the cell-free supernatant (CFS) of SNBF-1 enhanced glucose consumption and glycogen synthesis in insulin-resistant HepG2 cells, indicating improved cellular energy metabolism. This was consistent with the observation that the CFS of SNBF-1 increased the proliferation of HepG2 cells by 123.77 ± 0.82% compared to that of the control. Overall, this research significantly enhances our understanding of NGPs and their potential therapeutic applications in modulating the gut microbiome.

## 1. Introduction

The human gut microbiota, consisting of a sophisticated assembly of microorganisms, including bacteria, fungi, and other entities, plays a crucial role in overall health [1]. It influences diverse facets, including nutrient absorption, immunity, and behavioral patterns [2]. Traditional probiotics like Lactobacillus and Bifidobacterium are well recognized for their positive effects on the well-being of their host. However, recent advances in molecular biology and an enhanced understanding of the gut microbiome have led to a paradigm shift towards next-generation probiotics (NGPs) [3,4]. Unlike their predecessors, NGPs are celebrated for their specialized therapeutic properties and are selected for targeted functionalities within the gut ecosystem [5,6].

Although traditional probiotics offer a range of health benefits, their capacity to target specific diseases is frequently limited, and their efficacy varies considerably across different strains [7]. Traditional probiotics, often linked to fermented foods and not originally classified as medicinal products, faced scrutiny for their stability in the gut and limited immunomodulatory effects. This has led to the pursuit of NGPs which, derived from novel sources such as the human microbiome, offer targeted benefits and detailed safety profiles. Unlike the empirical development of traditional strains like *Lactobacillus* and *Bifidobacterium*, NGPs are identified through rigorous screening, showing promise for personalized therapy despite potential risks that necessitate a thorough safety evaluation. The move toward NGPs represents a response to the shortcomings of traditional probiotics, aiming for more precise and effective treatments [8,9]. Notably, non-toxigenic *B. fragilis* stands out for its role in modulating immunity and reducing inflammation [10,11]. For instance, Strain NCTC 9343 is instrumental in the prevention of chronic colitis and tumor development, operating without reliance on the polysaccharide A molecule. By outcompeting the pathogenic enterotoxigenic *B. fragilis*, reducing levels of the inflammatory marker IL-17A, and employing mechanisms of competitive exclusion, it effectively suppresses colitis. Nonetheless, the efficacy of NCTC 9343 is primarily preventative, offering limited therapeutic benefits once colitis is already established [12]. Research on the probiotic functions of NTBF, especially in domestic environments, is still in its early stages. By focusing on *B. fragilis* strains, we highlight their unique characteristics and potential applications. As crucial gut microbiota members, they are vital in maintaining gut health and influencing disease [13,14]. These strains provide numerous health benefits, including modulating the immune system [11,15], protecting against gastrointestinal diseases, and showing potential to manage metabolic disorders [10,16]. The specific attributes of *B. fragilis*, particularly those enhancing their resilience and adaptability in the gut environment, are of significant scientific interest. Understanding these attributes deepens our insight into gut microbiome dynamics and paves the way for novel therapeutic strategies.

This study aims to comprehensively analyze the cultivation methods and functional properties of the *B. fragilis* strain named SNBF-1 (Figure 1B). The examination of growth conditions, viability, and therapeutic potential is intended to advance research into the gut microbiome and its implications for health management. The focus is on optimizing the growth of *B. fragilis* strains and assessing their therapeutic efficacy in the human gut. Research on *B. fragilis*, representative of NGPs, illuminates their potential in advancing personalized probiotic therapy and gut microbiome modulation.

## 2. Materials and Methods

### 2.1. Microorganism Isolation

SNBF-1 was isolated from the feces of 1–3-month-old infants in Heilongjiang, Liaoning, and Jiangsu, China. The strain was cataloged in the China General Microbiological Culture Collection Center (CGMCC) under the accession number, i.e., CGMCC number, of 28586 on 1 October 2023. Fecal samples were thawed and diluted in sterile saline, creating dilutions from 10^−1^ to 10^−7^. Subsequently, 100 µL of each sample was spread on Bacteroides Bile Esculin (BBE) [17] and Laked Vancomycin (LKV) agar plates and incubated at 37 °C for 48 h in an anaerobic chamber. From the distinct colonies that emerged, select colonies were streaked onto new BBE and LKV plates, followed by a repeat of the incubation process. The purified strains were then enriched in an optimized Brain Heart Infusion (BHI) liquid medium, Gram-stained, and examined under a microscope. Finally, after preservation in glycerol, they were stored at −80 °C, resulting in the isolation of 56 *B. fragilis* strains.

#### 2.1.1. Genetic Analysis of Carbohydrate Metabolism for Enhanced Sugar Utilization and Antibiotic Resistance in Optimized Culture Medium

Genomes of *B. fragilis*, downloaded from the NCBI microbial genome database, were annotated for enzymes and genes related to carbohydrate metabolism. We utilized the Carbohydrate Active Enzyme Database (CAZy, http://www.cazy.org/, accessed on 8 August 2022), Comprehensive Antibiotic Research Database (CARD, https://card.mcmaster.ca), and Kyoto Encyclopedia of Genes and Genomes Database (KEGG) for this annotation, accessed on 8 August 2022. All strains used in the study were sourced from the human gut. The data were visualized using R with the ggplot2, BiocManager, and GenomicFeatures packages.

#### 2.1.2. Bile Salt Hydrolase Activity and Survival in Simulated Gastrointestinal Conditions

The tolerance of the *B. fragilis* strain to acid and bile salts was assessed. Acidic environment tolerance was evaluated at pH 2.0, 3.0, and 4.0, observing increased survival rates with rising pH levels. Bile salt tolerance was examined in a BHI medium supplemented with varying bile salts, mimicking the stomach’s food retention duration.

Bile Salt Hydrolase Activity: Fresh bacterial cultures were streaked in triplicate on MRSc agar plates containing 0.37 g/L CaCl_2_ and 5 g/L sodium taurodeoxycholic acid. These plates were incubated aerobically at 37 °C for 72 h. Bile salt precipitation around the colonies indicated a positive bile salt hydrolase activity.

Strain resilience in artificial gastrointestinal and intestinal fluids was also investigated, calculating survival rates as the ratio of viable cell count after 4 h relative to the initial count. Survival rate was determined using the formula: Survival rate (%) = (A2/A1 × N0) × 100%, where A1 is the initial viable bacteria count in artificial intestinal fluid (CFU/mL), and A2 is the count after 4 h.

#### 2.1.3. Cell Surface Hydrophobicity

We conducted a cell surface hydrophobicity assay using xylene, following the methodology reported before [18]. The apolar solvent was used to assess the hydrophobicity and hydrophilicity of the SNBF-1 strain. To prepare the samples, the SNBF-1 strain was centrifuged at 4500 rpm for 10 min at room temperature, washed twice with sterile PBS, and resuspended to achieve an optical density (A600) of 0.6 ± 0.05. We measured the absorbance (A) of the upper layer of the bacterial suspension at 600 nm at 2, 4, 6, 12, and 24 h intervals, taking triplicate readings at each time point to ensure reliability and accuracy.

#### 2.1.4. Auto-Aggregation Assay

For the auto-aggregation assay, bacterial isolates were cultured as previously detailed and standardized to an optical density of 0.60 ± 0.05 (A600 nm), equivalent to 10^7^–10^8^ CFU/mL [19]. The bacterial suspension was then incubated at 15 °C, and the absorbance of its upper portion was measured hourly for 5 h. We calculated the auto-aggregation percentage using the formula:

Auto-aggregation ability (%) = [1 − (A600 at time T/A600 at time 0)] × 100%. Absorbance readings were taken at 0, 2, 4, 6, 12, and 24 h (T) to determine the auto-aggregation percentage over time.

#### 2.1.5. Enterotoxin PCR and Hemolysis Tests

Activated *B. fragilis* strains were incubated overnight in a liquid medium for DNA extraction. Polymerase Chain Reaction (PCR) amplification used enterotoxin gene primers (forward: 5′−GATGCTCCAGTTACAGCTTCCATTG−3′, reverse: 5′−CGCCCAGTATATGACCTAGTTCGTG−3′), targeting a 970 bp fragment. Post-PCR, the products underwent nucleic acid electrophoresis in 2% agarose gel at 100 V for 15 min.

The activated *B. fragilis* strains were streaked onto Brucella agar blood plates. These plates were then incubated at 37 °C for 48 h. The supernatant was taken to determine the OD570 nm value. After incubation, the colonies were observed for the presence or absence of a hemolysis zone around them.

### 2.2. Characterization of B. fragilis

#### 2.2.1. Growth Characteristics

The growth curve of the *B. fragilis* strain SNBF-1 was determined by inoculating the activated strain into a refined BHI liquid medium and incubating it anaerobically at 37 °C for 24 h. The growth cycle was monitored by measuring the optical density at 600 nm (OD600 nm) every 2 h.

#### 2.2.2. Preparation of Samples

Cell-free supernatant (CFS): The sample was cultured until reaching 10^3^–10^9^ CFU/mL, then centrifuged at 5000 r/min for 15 min at 4 °C to collect the supernatant. The supernatant was filtered through a 0.22 µm syringe filter to prepare the strain supernatant.

Intracellular cell-free extract (CFE): Cells at a concentration of 10^9^ CFU/mL were disrupted using an ultrasonic processor (KQ-500TDE, Kunshan Ultrasonic Instruments Co., Ltd., Kunshan, China). The sonication settings were 240 W for 10 min, performed in an ice bath to mitigate overheating. After sonication, the cell debris was removed by centrifugation at 5000 r/min for 15 min at 4 °C. The clear supernatant obtained after centrifugation was collected and used as the CFE.

#### 2.2.3. Cholesterol Removal Rate

We add cholic acid (3 mg/mL) and cholesterol (0.1 mg/mL) to create a high-cholesterol BHI culture medium. Activated strains are then inoculated into this medium and anaerobically incubated at 37 °C for 16 h. Post-incubation, the cholesterol content in the supernatant is quantified using the ortho-phthalaldehyde method. Cholesterol removal rate (%) = (m_1_ − m_2_)/m_1_ × 100% (where m_1_ and m_2_ represent the mass of cholesterol in the supernatant before and after fermentation, respectively, in µg).

#### 2.2.4. Antibiotic Resistance Evaluation

In the Kirby–Bauer disc agar diffusion test (K–B), the drug sensitivity of SNBF-01 was assessed. A 100 μL aliquot of the 1.0 × 10^9^ CFU/mL activated SNBF-1 suspension was uniformly spread on the BHI agar medium. Drug-sensitive papers, as listed in Table 1 were then placed on the agar and left for 30 min. Subsequently, the setup was incubated at 37 °C for 24 h to measure the diameter of the inhibition zones. The assessment of antibiotic resistance was based on the guidelines provided by the American Clinical Laboratory Standards Institute (CLSI).

#### 2.2.5. Antioxidant Assay:

##### DPPH Radical (DPPH•) Scavenging Effect

The ability of SNBF to clear DPPH• radicals was assessed using previously described methods [20] with minor modifications. Briefly, 1.0 mL CFS or CFE solution was mixed with 0.2 mL DPPH• solution (0.4 mM) and 2.0 mL deionized water. The absorbance of the solution was recorded at 517 nm. The clear DPPH• was calculated using the following equation:$$\mathrm{DPPH}\;\mathrm{Scavenging}\;\mathrm{Effect}\;(\%)=(\frac{Absorbance\;of\;control-Absorbance\;of\;sample}{Absorbance\;of\;control})\times 100\%$$

##### Hydroxyl Radical (•OH) Scavenging Activity

Briefly, 1 mL CFS or CFE was mixed with 40 μL of 9.0 mM FeSO4, 40 μL of 0.03% H_2_O_2_, and 20 μL of a 9.0 mM salicylic acid–ethanol solution, then incubated at 37 °C for 30 min. The absorbance change caused by salicylic acid was measured at 510 nm. Furthermore, 2 mg/m ascorbic acid (Vc) was used as a reference material. The hydroxyl radical scavenging activity was calculated using the following formula:

Scavenging activity (%) $=(\frac{As-A0}{Ac-A0})\times 100$%As: OD value of the sample with CPS.A0: OD value of the blank (distilled water instead of CPS).Ac: OD value of the control (no H_2_O_2_).

##### ABTS Radical (ABTS•^+^) Scavenging Activity

The antioxidant activity of pure cell-free supernatant (CFS) can be evaluated using the ABTS (2,2′-azino-bis(3-ethylbenzothiazoline-6-sulfonic acid) clearance test [21]. A 1 mL volume of CFS samples was combined with the ABTS•^+^ working solution. These mixtures were incubated at 37 °C for 8 min in the dark, and their absorbance was measured at 734 nm. The percentage inhibition of absorbance was calculated against a Trolox standard curve to determine the scavenging activity.

##### Oxygen Radical (O_2_ •) Absorbance Capacity Assay

O_2_ • scavenging capacity was determined by the oxygen radical absorbance capacity (ORAC) method [22]. Briefly, for each assay, 200 μL of fluorescein sodium solution was added to a 96-well plate, followed by 20 μL of the CFS sample. The mixture was shaken for a minute and then incubated at 37 °C for 10 min. After adding 20 μL of 2,2′-azobis(amidinopropane) dihydrochloride, fluorescence was measured every minute with an excitation of 485 nm and emission of 535 nm. Glutathione (GSH) served as the positive control. The fluorescence’s area under the curve (AUC) was calculated using the formula:AUC=0.5×∑(fn+fn+1)×Δt

*f_n_* is the relative fluorescence intensity at the *n*-th measurement point, and Δ*t* is the time interval between consecutive points.

The ORAC value, expressed in µM *Trolox* equivalents per mg of sample, was determined by:$$ORAC\;value=\frac{AUCSample-AUCAAPH}{molarity\;of\;Trolox}\times \frac{AUCTrolox-AUCAAPH}{molarity\;of\;Trolox}$$

Here, AUCSample is the fluorescence area under the curve with antioxidants, AUCAAPH is the area under the curve with radical action in the absence of antioxidants, and AUCTrolox is the area under the curve with standard antioxidants. Molarities for *Trolox* and the sample are provided in μmol/g and mg/mL, respectively. This method quantifies the antioxidant capacity of the sample compared to that of *Trolox*.

### 2.3. In Vitro Investigation of Lipid Accumulation Reduction by B. fragilis Strain SNBF-1

#### 2.3.1. Culture and Treatment of HepG2 Cells

The HepG2 cells, obtained from Beijing Solarbio Science & Technology Co., Ltd. (Beijing, China), were grown in 100 cm² Petri dishes with DMEM containing 10% fetal bovine serum and antibiotics. They were kept at 37 °C in a 5% CO_2_ environment. The study used a PBS buffer with NaCl, KCl, Na_2_HPO_4_, and KH_2_PO_4_ in distilled water, adjusted to a pH of 7.2–7.4.

#### 2.3.2. MTT Viability Assay

HepG2 cells were seeded in 96-well plates at 2 × 10^5^ cells/mL and cultured for 24 h. Post-culture, cells were washed with PBS and incubated with 100 μL of sample for another 24 h. Then, 20 μL of 0.5 mg/mL MTT solution was added to each well, followed by 4 h of incubation at 37 °C in darkness. Afterward, the supernatant was discarded, 150 μL of DMSO was added, and the mixture was shaken for 10 min. Absorbance was measured at 490 nm using a SpectraMaxM2e microplate reader. Cell viability results were expressed as a percentage relative to the control, assumed to be 100% viable.

#### 2.3.3. Oil Red O Stains Intracellular Lipids

HepG2 cells were cultured in a 6-well plate (2 × 10^5^ cells per well) for 24 h, washed with PBS, and the supernatant was discarded. Groups included blank, model, CFS, CFE, and positive control (Simvastatin). Each group, except the blank, received specific treatments and was induced with 0.25 mmol/L palmitic acid for 24 h. Afterward, cells were washed with PBS, fixed with 10% neutral formaldehyde, and stained with Oil Red O for 30 min. Cells were then washed with isopropanol and ethanol, rinsed with water, and air-dried. Finally, they were counterstained with hematoxylin for 15 min, washed, and readied for observation and photography.

#### 2.3.4. Lipid Accumulation, TC, TG, HDL-C, and LDL-C Assays

HepG2 cells were seeded at 2 × 10^5^ cells/well in 6-well plates, incubated with samples, and subjected to co-incubation with oleic acid and palmitic acid. Following incubation, cells were stained for lipid analysis or lysed for total cholesterol (TC), triglycerides (TG), HDL-cholesterol (HDL-C), and LDL-cholesterol (LDL-C) assays. These assays were performed using assay kits (Nanjing Jiancheng).

### 2.4. In Vitro Investigation of Glucose Metabolism by B. fragilis Strain SNBF-1

#### 2.4.1. Inhibition of α-Amylase and α-Glucosidase Activity

In the α-amylase assay [23], 500 μL of CFS and CFE from isolates were mixed with an equal volume of 0.1 M PBS containing α-amylase (0.5 mg/mL, pH 6.4) and incubated at 25 °C for 10 min. Next, 500 μL of 1% starch in 0.1 M PBS (pH 7.4) was added, followed by a further 10 min incubation at the same temperature. The reaction was stopped with 1 mL of DNS reagent and heated in a boiling water bath for 5 min. After cooling to room temperature and diluting with 10 mL of water, the absorbance was measured at 540 nm to calculate α-amylase inhibition:$$inhibiting\;effect(\%)=[\frac{1-A540\left(Sample\right)}{A540\left(Control\right)}]\times 100\%.$$
where A540 (Control) is the absorbance at 540 nm of the control sample without protein extract, and A540 (Sample) is the absorbance at 540 nm of the test sample.

In the α-glucosidase assay [24], yeast α-glucosidase (100 U/mg) was used. Test samples (CFS and CFE100 μL) were mixed with 50 mM PBS (pH 6.8) and incubated for 10 min. α-glucosidase (100 μL, 0.25 U/mL) was then added and pre-incubated at 37 °C for 15 min. This was followed by the addition of 100 μL of 5 mM pNPG and a further 30 min incubation at 37 °C. The reaction was stopped with 1000 μL of 0.1 M Na_2_CO_3_, and the absorbance of 4-nitrophenol was measured at 405 nm to calculate percent inhibition.
$$inhibiting\;effect(\%)=[\frac{1-A405\left(Sample\right)}{A405\left(Control\right)}]\times 100\%.$$
where A405 (Control) is the absorbance at 540 nm of the control sample without protein extract, and A405 (Sample) is the absorbance at 540 nm of the test sample.

#### 2.4.2. The IR-HepG2 Cell Model for Antidiabetic Test

To establish an insulin resistance (IR) model, 2 × 10^5^ HepG2 cells/well were cultured until adherence and induced with insulin-containing DMEM for 48 h [25,26]. Cultured HepG2 cells were divided into groups to assess the influence of CFS and CFE on glycogen synthesis, which was determined using a glycogen assay kit following the manufacturer’s instructions (Nanjing Jiancheng Bioengineering Institution). The activities of hexokinase (HK) and pyruvate kinase (PK) in IR-HepG2 cells were measured with specific assay kits. The inhibitory effects of various strain supernatants on α-amylase and α-glucosidase were assessed. Changes in glucose levels in IR-HepG2 cells were monitored post-supernatant treatment. Protein extracts and Acarbose were prepared in water. Porcine α-amylase inhibition was measured with dinitrosalicylic acid. After pre-incubation and starch addition, reactions were stopped and diluted, and their absorbance at 540 nm was measured to calculate percentage inhibition.

### 2.5. Statistical Analysis

Three parallels (*n* = 3) were set up for each of the above experiments, and the statistical analysis of the experimental data was performed using R studio and Prism 10. Observed differences were analyzed using one-way ANOVA and Duncan’s test at the level of 0.05.

## 3. Results

### 3.1. Identification and Characterization of B. fragilis Isolated from Fecal Samples

#### 3.1.1. Genetic Analysis of Carbohydrate Metabolism for Enhanced Sugar Utilization and Antibiotic Resistance in Optimized Culture Medium

To evaluate the growth characteristics of *B. fragilis* and optimize the culture medium as well as the isolation and screening process on a genomic scale, we annotated the genomes of 23 *B. fragilis* strains from GeneBank across the Carbohydrate Active Enzyme Database and Kyoto Encyclopedia of Genes and Genomes Database. This analysis allowed us to assess the correlation between genotype and phenotype concerning carbohydrate metabolism, providing insights into the medium’s suitability for cultivating these bacteria from the human intestine [27]. The results presented in Figure 1A reveal that, in comparison to all annotated *B. fragilis*, genes from families GH2, GH3, CH20, GH29, and GH92 are predominantly present in the *B. fragilis* genomes sourced from NCBI.

The Glycoside Hydrolase Family 29 (GH29) includes enzymes like α-L-fucosidases and α-L-galactosidases that process fucose and galactose sugars, targeting α-linked fucosyl residues and galactose in glycoproteins and glycolipids. The Glycoside Hydrolase Family 20 (GH20) contains β-N-acetylhexosaminidases, focusing on N-acetylglucosamine substrates and N-glycans in glycoproteins. Glycoside Hydrolase Family 3 (GH3) encompasses enzymes like β-glucosidases and β-xylosidases, crucial for breaking down complex carbohydrates. Collectively, these families play vital roles in carbohydrate metabolism. Regarding antibiotic resistance (Table 2), most *B. fragilis* strains carry the adeF and vanT genes. The adeF gene confers resistance to fluoroquinolones and tetracyclines through an efflux mechanism, while the vanT gene, part of the vanG cluster, provides resistance to glycopeptide antibiotics like vancomycin by modifying the bacterial target site.

For the final approach in isolating *Bacteroides fragilis* from fecal samples, we employed a BEE-based medium supplemented with 20 mg/L of tetracycline and 15 mg/L vancomycin. Additionally, starch was added as a carbon source to facilitate the selective growth of *Bacteroides fragilis* derived from human feces.

To examine the effect of various carbon, nitrogen, and phosphate sources on bacterial growth, the BHI medium was adapted by incorporating diverse carbon sources based on the genetic analysis of carbohydrate metabolism (refer to Figure 2). These carbon sources included glucose, sucrose, lactose, inulin, chitosan oligosaccharide, isomalto-oligosaccharide, soluble starch, fructooligosaccharide, xylooligosaccharide, galactooligosaccharide, and resistant dextrin. For nitrogen sources, we used tryptone, peptone, casein peptone, soy peptone, beef liver infusion, beef extract, and yeast extract. Phosphate sources comprised disodium hydrogen phosphate, sodium dihydrogen phosphate, dipotassium hydrogen phosphate, and potassium dihydrogen phosphate. Our results (see Figure 2) indicated that in the cultivation of SNBF-1 based on BHI, 0.5% soluble starch significantly increased the OD600 nm to 1.011, surpassing other carbon sources. In contrast, 2% soy peptone as a nitrogen source yielded an OD600 nm of 0.938. Additionally, 0.3% dipotassium hydrogen phosphate as a phosphorus source achieved an OD600 nm of 1.009, highlighting the critical role of nutrient composition in bacterial culture optimization.

#### 3.1.2. Acid and Bile Tolerance Tests and Artificial Gastrointestinal and Intestinal Fluids

The scatter plot (Figure 3A) presents the principal component analysis (PCA) results of the survival rates of 38 microbes at pH 2.5 and 1% of bile salts and in artificial gastric fluid (AGF) and artificial intestinal fluid (AIF). Each point on the plot represents a microbe, positioned based on the main variations in survival rates under the four conditions. Principal component 1 (PC1) accounts for approximately 62.05% of the data variance, indicating it captures the most significant source of variation in microbial survival rates. The distribution along PC1 likely reflects the microbes’ overall sensitivity or resistance to AGF and AIF conditions. For instance, microbes with higher PC1 values may exhibit higher survival rates in one or both conditions, whereas those with lower PC1 values may have lower survival rates. Notably, some microbes (Strains SNBF-1, CD11-1, CD11-2, CD11-5, CD13-1, CD13-4, and SY-X-3, those on the right side of the chart) may be more tolerant to these environmental conditions, while others are less tolerant. SNBF-1, in particular, shows higher values along PC1, suggesting it may have a higher survival rate in these conditions. This dominance in data variability by PC1 implies SNBF-1’s better adaptability or tolerance compared to other microbes under simulated digestive system conditions. SNBF-1’s high survival rate and adaptability could make it a valuable candidate for developing probiotic products, especially for applications requiring tolerance to harsh gastrointestinal conditions. For example, in food supplements, functional foods, or therapeutic probiotic products, microbes with high survival rates could more effectively reach the intestines and convey their benefits. The proportion of variance chart (Figure 3B) helps to decide how many principal components should be retained for an efficient yet comprehensive data representation. PC1 explains 62.05% of the variance, and PC2 explains an additional 19.06% of the variance. The fact that PC1 and PC2 together account for over 80% of the variance suggests that these two components capture most of the information in the original dataset and might be sufficient for retention.

Seven isolates (SNBF-1, CD11-1, CD11-2, CD11-5, CD13-1, CD13-4, SY-X-3) were selected from the 38 screened strains, each demonstrating higher PC scores. Detailed results are presented in Table 3. Notably, within the *B. fragilis* group, SNBF-1 showed remarkable tolerance to both acid and bile salts, with in vitro tolerance rates of 94.21 ± 1.12% for gastric fluids and 155.96 ± 1.55% for intestinal fluids.

#### 3.1.3. Cell Surface Hydrophobicity and Auto-Aggregation Assay

The cell surface hydrophobicity and the ability of strains to self-aggregate are related to their capacity to adhere and colonize the intestinal tract. For probiotic strains, a maximum of 40% cell surface hydrophobicity is necessary. The hydrophobicity and self-aggregation of the tested *B. fragilis* strains are shown in Table 4. We tested 38 non-toxigenic *B. fragilis* strains, revealing varying degrees of hydrophobicity. Strains HC-LX-1, SNBF-1, CD4-1, CD11-1, CD11-2, CD13-4, SY-XB-1, SY-X-2, and JS1-4 exhibited moderate hydrophobicity, while the others were weakly hydrophobic. The strongest hydrophobicity was observed in strain CD13-4 (51.48 ± 0.95%). After 16 h of static incubation, most strains demonstrated high self-aggregation ability, exceeding 60%, with strains CD4-1, CD11-2, 1, CD13-4, and SY-X-3 reaching 65.84 ± 3.19%, 52.88 ± 4.09%, 66.52 ± 6.25%,77.79 ± 7.11%, and 70.54 ± 2.88%, respectively.

### 3.2. Characterization of B. fragilis

#### 3.2.1. Cholesterol Removal Rate and Bile Salt Hydrolase Activity

The diamond-shaped data points in light green color correspond to the cholesterol removal rates of each strain, with SNBF-1 outperforming all strains by reducing cholesterol by 55.38% ± 2.26% (Figure 4). The robust abilities of strains CD11-1 and CD13-4 are also evident, reducing cholesterol by 49.99% ± 1.60% and 48.46% ± 1.53%, respectively. In contrast, SY-X-3 is shown to have the lowest reduction rate at 22.08% ± 1.04%. Additionally, the bile salt hydrolase activities are represented by light red dots, where SNBF-1 shows the highest activity at 410.04 U/L ± 12.29. Strains CD11-1, CD13-1, and CD13-4 display similar high enzyme activities, whereas SY-X-3 lags significantly behind, making it the least active strain. This figure underscores the diverse capabilities of *B. fragilis* strains in cholesterol reduction and bile salt hydrolase activity within the study. Collectively, these findings provide valuable insights into the potential use of these strains in therapeutic applications for cholesterol-related health conditions.

#### 3.2.2. Antibiotic Resistance Evaluation

The tested strains’ response to different antibiotics is summarized in Table 2. Notably, the strains exhibit resistance to Ampicillin, Polymyxin E, Ciprofloxacin, and Sulfadiazine, as indicated by the ‘R’ entries in the corresponding columns for these antibiotics. Specifically, strains SNBF-1 and CD11-1 showcase resistance to all four of these antibiotics. Additionally, the remaining strains, CD11-2, CD11-5, CD13-1, CD13-4, and SY-X-3, exhibit resistance to more than four antibiotics, denoted by the ‘R’ entries in their respective columns. However, it is important to observe that the degree of resistance may vary among the strains, suggesting distinct sensitivity patterns to the tested antibiotics.

#### 3.2.3. Antioxidant Assay

In Figure 5A, CD11-1’s CFS shows the highest DPPH scavenging activity at 82.50%, surpassing its CFE at 77.21%, whereas CD11-5 exhibits the lowest activity for both CFS (51.20%) and CFE (48.35%). The results from Figure 5B indicate CD13-4’s CFS as the most effective in hydroxyl radical scavenging at 80.52%, with SY-X-3’s as the least effective at 39.31%. Figure 5C’s ABTS assay findings are consistent, with CFS outperforming CFE across the strains. Finally, Figure 5D shows CD11-1’s CFS with the highest reducing power (1.368 absorbance), in contrast to CD11-5’s CFE, which has the lowest (0.517 absorbance), reaffirming that CFS typically exhibits superior antioxidant activity compared to CFE.

#### 3.2.4. MTT Viability Assay

We assessed the impact of various concentrations of bacterial supernatant and cell-free extracts (ranging from 10^3^ to 10^9^ CFU/mL) on the viability of HepG2 cells, employing the different concentrations of supernatant and cell-free extracts to interact with the cells for 24 h. The results, as depicted in Figure 6A, indicate that at a concentration of 10^9^ CFU/mL of bacterial supernatant, there was a significant decrease in HepG2 cell viability compared to that of the control group. At concentrations of 8 × 10^8^ CFU/mL, 6 × 10^8^ CFU/mL, 4 × 10^8^ CFU/mL, and 2 × 10^8^ CFU/mL, the four groups of supernatants did not show a significant difference in the survival rate of HepG2 cells compared to that of the control group. The cell-free extracts of the bacterial strain demonstrated a significant increase and no significant difference in the survival rate of HepG2 cells compared to that of the control group. Consequently, CFS at concentrations of 8 × 10^8^, 6 × 10^8^, and 4 × 10^8^ CFU/mL, along with CFE at 10^9^, 10^8^, and 10^7^ CFU/mL, were selected for subsequent experiments.

#### 3.2.5. Oil Red O Stains Intracellular Lipids

SNBF-1 ability to inhibit lipid accumulation was tested on HepG2 cells exposed to oleic acids (OA), inducing steatosis. After staining with Oil Red O and hematoxylin, cells showed increased lipid accumulation and shape changes, confirming steatosis induction. However, co-treatment with CFS and CFE significantly reduced this lipid accumulation and helped maintain normal cell morphology, as shown in Figure 6B,C. The reduction in triglycerides (TG) and low-density lipoprotein cholesterol (LDL-C) further confirmed SNBF-1’s protective effects against fatty acid-induced lipid accumulation.

### 3.3. Lipid Accumulation, TG, and LDL-C Assays

The effects of CFS and CFE on lipid accumulation and TC content in cells are shown in Figure 7. Compared to the blank group, the model group showed a significant increase in TC content (*p* < 0.05). Both the CFS and CFE groups, along with the positive control group (treated with Simvastatin), showed a significant decrease in TC content compared to that of the model group, indicating practical lipid-lowering effects. The positive control had the lowest TC content and the most significant lipid-lowering impact. There was no significant difference between the CFS and CFE groups, both significantly reducing TC content in lipid-accumulating cells.

Furthermore, Figure 7 shows that the model group had a significant increase in triglycerides (TG) content (*p* < 0.05) compared to that of the blank group. After intervention with CFS, CFE, and Simvastatin, the TG content in lipid-accumulating cells significantly decreased. The bacterial supernatant group’s TG content was comparable to that of the positive control, indicating a strong ability to reduce TG and overall lipid levels.

The model group’s high-density lipoprotein cholesterol (HDL-C) content significantly decreased after the addition of palmitic acid to 21.07 ± 1.08%, compared to the blank group’s higher level of 81.09 ± 0.79% (*p* < 0.05). However, the positive control group, treated with Simvastatin, showed a substantial increase in HDL-C content to 77.39 ± 1.43%. The CFE group’s HDL-C content was similar to the model group at 22.00 ± 0.62%, indicating no significant change, while the CFS group showed a slight increase to 27.72 ± 0.75%, although the effect was modest. These findings suggest that the bacterial supernatant has a limited effect on raising HDL-C levels, and the cell-free extract does not appear to enhance HDL-C.

For low-density lipoprotein cholesterol (LDL-C), the model group experienced a significant increase to 118.65 ± 0.59% following palmitic acid addition (*p* < 0.05). Post-treatment, LDL-C levels were significantly reduced in the CFS group to 62.69 ± 0.65%, in the CFE group to 66.60 ± 1.08%, and in the positive control group to 23.94 ± 0.57%, with Simvastatin showing the most substantial reduction. Although the effect of the bacterial supernatant on lowering LDL-C was not as pronounced as that of Simvastatin, it still showcased a notable lipid-lowering capability.

### 3.4. In Vitro Investigation of Glucose Metabolism by B. fragilis Strain SNBF-1

Figure 8 shows the in vitro investigation of glucose metabolism by *B. fragilis* Strain SNBF-1. The inhibition rate of strain SNBF-1 on α-amylase reached 87.04 ± 2.03%. SNBF-1 exhibited the highest inhibition rate on α-glucosidase, reaching 37.82 ± 1.36%. Our study revealed that, after insulin induction, the model group’s glucose consumption was significantly decreased, confirming insulin resistance, with an average of 421.56 ± 6.45%, compared to the blank group that had a notably higher consumption at 960.64 ± 26.67% (*p* < 0.05). Contrastingly, treatments with CFS and CFE improved glucose consumption to 672.22 ± 10.83% and 606.41 ± 11.07%, respectively, which suggests their potential in alleviating the IR-HepG2 cell state by increasing glucose uptake.

For glycogen content between the model group and the blank group, the model control exhibited a significantly reduced glycogen content of 35.60 ± 0.32%, compared to the blank group’s 87.13 ± 0.33% (*p* < 0.05). However, treatments with CFS and CFE increased the glycogen content in the IR-HepG2 cells to 54.44 ± 0.38% and 51.98 ± 0.36%, respectively, indicating their effectiveness in enhancing glycogen synthesis and thereby ameliorating insulin resistance. Metformin, serving as the positive control, demonstrated the most substantial increase in glycogen content, reaching 67.91 ± 0.28%, and hence showed the strongest effect among the interventions. Although not as potent as metformin, the increase in glycogen levels attributable to CFS and CFE still represents a significant improvement in the cellular state of insulin resistance, with CFS having a marginally superior efficacy over CFE.

Activities of HK and PK were markedly reduced in the model group (*p* < 0.05). After intervention with CFS, CFE, and metformin, enzyme activities increased, facilitating glucose absorption and metabolism. Metformin significantly improved HK and PK activities in IR-HepG2 cells compared to those of the model group. Post-intervention, both HK and PK activities increased with CFS and CFE treatments, where CFS had a better effect than CFE.

## 4. Discussion

*B. fragilis*, a pivotal inhabitant of the human gastrointestinal tract, plays a dual role in health, acting both as a beneficial probiotic and a potential opportunistic pathogen [11]. Early research has revealed that *B. fragilis* plays a role in modulating the gut microbiome by interfering with the growth or translocation of other microbes. In animal models, *B. fragilis* has been evaluated for its potential in preventing *Clostridioides difficile* infection (CDI) [28]. This was performed by supplementing mice prophylactically with *B. fragilis*, which led to improved bacterial diversity and a positive correlation with the abundance of *Akkermansia muciniphila*. *B. fragilis* was shown to inhibit C. difficile adherence by preventing apoptosis and the loss of zonula occludens-1 (ZO-1) and mucin-2 (MUC-2) [29]. In another investigation, *B. fragilis* cultures were found to inhibit the translocation of Salmonella Heidelberg, attributed to the secretion of antimicrobial protein-1 (BSAP-1). This protein, containing membrane attack/perforin (MACPF) domains, is capable of lysing bacterial cells or infecting host cells, indicating its therapeutic potential in enhancing gut health and preventing infections. For other mechanisms in the field of gut microbiota competition, *B. fragilis*’s type VI secretion system (T6SS), especially its GA3 loci, was found to antagonize most human gut Bacteroidales strains [30]. The GA3 T6SS in *B. fragilis* strain 638R, active in mammals, provides a competitive advantage by deploying unique toxins against rival bacteria, potentially carving out a protected niche in the human colon. This underexplored mechanism offers promising avenues for creating therapies aimed at regulating gut flora and addressing gastrointestinal health.

For immune regulation, studies note that *B. fragilis*’s Polysaccharide A (PSA) is crucial and may protect against diseases like bowel disease [11,31]. *B. fragilis* administration in animal models leads to the binding of PSA with B cells, crucial for inducing regulatory CD4+ and CD8+ T cells that secrete IL-10, thereby controlling innate inflammatory responses. This process results in decreased colonic levels of inflammatory cytokines (TNF-α, IL-1β, IL-6) and an increase in IL-10 [32], highlighting the anti-inflammatory effects of *B. fragilis*.

However, *B. fragilis* strains are not devoid of adverse attributes. In certain conditions, it can turn into an opportunistic pathogen, leading to severe infections [33,34]. Its link to autoimmune disorders and colorectal cancer, through the production of *B. fragilis* enterotoxin (BFT) and a protein similar to human ubiquitin [15,33,35], further adding to its complex nature. Building upon the known effects of BFT, new research highlights the role of B. fragilis and its metabolites, specifically 12-hydroxy-heptadecatrienoic acid (12-HHTrE) and Prostaglandin E2 (PGE2), in neurological health. These substances have been found to activate microglia, contributing to the pathogenesis of Alzheimer’s disease (AD) in neuronal C/EBPbeta transgenic mice [10].

Understanding *B. fragilis*’s behavior within the gut is crucial. While it can bolster our gut health and ward off harmful pathogens, its capacity for antibiotic resistance and toxin production poses significant challenges [36]. Selecting strains without pathogenic potential is vital to reduce these risks. Additionally, understanding its role in lipid metabolism is important for broader health perspectives. The study of *B. fragilis* is essential for developing effective treatments and maintaining gastrointestinal health. By continuing to explore its multifaceted role, we can better leverage its benefits while minimizing the adverse effects, ensuring a healthier gut ecosystem.

From 43 fecal samples collected from different regions, 79 strains of suspected *B. fragilis* were isolated. After conducting a 16S rDNA sequence homology analysis on these 79 strains, a total of 56 strains of *B. fragilis* were identified. Subsequently, we tested these for hemolysis. Among all the *B. fragilis* strains screened, 11 showed positive results for the expression of enterotoxin genes. In total, 38 exhibited gamma-hemolysis. Hemolysis tests in microbiology are essential for distinguishing and identifying specific bacterial pathogens. These tests are carried out on blood agar plates (BAP), which serve both as enriched and differential media in clinical environments. The blood in these agar plates provides essential nutrients for ‘fastidious’ bacteria, which need specific or additional types of nutritional support. The observed hemolysis patterns aid in differentiating various bacterial species.

The interaction of bacteria with the intestinal environment is significantly influenced by their hydrophobic nature and auto-aggregation ability, playing crucial roles in their ability to colonize the gut and exert beneficial effects [37,38]. Hydrophobic interactions enable bacteria to adhere more effectively to each other and to the mucosal surfaces, thereby enhancing their auto-aggregation. This synergistic relationship not only strengthens their adhesion to the intestinal mucosa, which is inherently hydrophobic due to mucus layers, but also facilitates an expanded habitat for beneficial bacteria, such as Lactobacillus plantarum, thus contributing to a balanced gut microbiota. These properties are vital for the stable functioning of probiotics in the gut, enhancing their resistance to pathogens through competitive exclusion, supporting biofilm formation to resist washout, and improving gastrointestinal transit tolerance. Together, these mechanisms highlight the critical roles of hydrophobicity and auto-aggregation in the effectiveness of probiotics and in maintaining a healthy gut microbiome by aiding in bacterial adhesion, colonization, and health benefit provision to the host.

Upon entering the human body, probiotics begin to exert their functions as they transit from the oral cavity to the intestinal tract. To be effective, probiotics must withstand the highly acidic environment of the stomach, where gastric pH can range from 1.3 to 1.8, and adapt to around 3.5 post-meal due to food neutralization and dilution [39]. Moreover, the ability to tolerate bile salts, which possess antibacterial properties, is essential for probiotics to adapt to the environment of the human small intestine and manifest beneficial effects [40]. The adhesion of probiotics to the intestinal mucosa is closely linked to their surface hydrophobicity and is enhanced by their capacity for auto-aggregation, especially during rapid growth phases. Strains that exhibit strong auto-aggregation have improved adhesion abilities, underscoring the importance of these characteristics in evaluating the resilience and functional efficacy of probiotic strains [41].

With the expanding use of broad-spectrum antibiotics in medicine, the antibiotic resistance of pathogenic strains intensifies, leading to imbalances in the gut microbiota [42]. In the realm of medical microbiology, the rising antibiotic resistance among Bacteroides fragilis strains presents a significant public health concern. *B. fragilis*, while constituting a minor component of the human colonic flora, emerges as a predominant Gram-negative anaerobic pathogen, particularly in immunocompromised individuals. This organism is implicated in a wide array of infections and produces a notable virulence factor known as fragilysin, which is associated with gastrointestinal pathologies, colorectal cancer, and inflammatory bowel diseases. A recent study aimed to elucidate the correlation between drug resistance and specific genetic markers in clinical *B. fragilis* strains [43]. Specifically, resistance rates to penicillin G, clindamycin, cefoxitin, and amoxicillin/clavulanic acid were notably high, while all strains remained susceptible to imipenem and metronidazole. Critical resistance genes such as cepA, cfxA, cfiA, and ermF were identified in varying proportions of the isolates. Notably, a strong correlation was established between cefoxitin resistance and the cfxA gene, as well as between clindamycin resistance and the ermF gene.

Therefore, determining the antibiotic sensitivity of probiotics is a primary task for their application in the food industry. In this study, we identify seven strains suitable for safe intestinal colonization, including SNBF-1, CD11-1, CD11-2, CD11-5, CD13-1, CD13-4, and SY-X-3. Their acid, bile salt, and gastrointestinal fluid tolerance are higher than those of some heterologously sourced strains reported in the literature, showing potential research value (Table 2).

Glycogen synthase activity, regulated by various factors, plays a pivotal role in glycogen synthesis and glucose metabolism [44]. The diminished activity of this enzyme disrupts glycogen synthesis, leading to metabolic imbalances. In insulin-resistant HepG2 cells, enhancing the activities of HK and PK may significantly improve glucose uptake, offering a promising strategy to counteract insulin resistance. HK, as the initial rate-limiting enzyme in glycolysis, is crucial for regulating glucose’s entry into this metabolic pathway. Concurrently, PK, instrumental in glycolysis’s final stages, is vital for cellular energy production. The diminished activities of these enzymes are linked with reduced glycogen synthesis and inefficient glucose processing, contributing to heightened blood glucose levels [45]. Therefore, modulating HK and PK activities emerges as a strategic approach for maintaining glucose homeostasis and combating insulin resistance.

Particularly in the context of insulin-resistant cells like IR-HepG2, HK and PK activities could significantly influence both anaerobic and aerobic glucose metabolism pathways. For example, increased HK activity enhances the initial step of glycolysis, ensuring more efficient glucose conversion into glucose-6-phosphate [46]. Similarly, elevated PK activity at the final stage of glycolysis can lead to more effective pyruvate production. This enhancement in enzyme activities is anticipated to foster improved glucose uptake and utilization, thus aiding in the management of blood glucose levels and offering therapeutic benefits for those grappling with insulin resistance.

For instance, enhancing HK and PK activities in IR-HepG2 cells might result in better control of glycemic levels in diabetic patients. Targeted interventions aimed at increasing these enzymes’ activities could potentially rectify the metabolic aberrations associated with insulin resistance. This presents a methodical approach to alleviate its overall impact on the body, such as reducing the risk of developing diabetes-related complications like neuropathy and nephropathy.

It is essential to acknowledge the inherent limitations that accompany our findings. This research, conducted exclusively in vitro, presents an initial exploration into the effects of the SNBF-1 strain. While these preliminary results are promising, it is crucial to recognize that in vitro conditions may not fully replicate the complex interactions and environmental variables present within the human body. Consequently, the extrapolation of these findings to in vivo contexts should be approached with caution. The enduring effects of this strain on human health, both beneficial and potentially adverse, are yet to be delineated. This gap in knowledge underscores the imperative need for extensive longitudinal studies.

Given these considerations, the urgency for further research is paramount. Future studies are critical for confirming and building upon preliminary results, as well as comprehensively assessing the safety and effectiveness of the SNBF-1 strain in clinical environments. Through detailed and broad research efforts, our goal is to enhance our understanding of *B. fragilis* SNBF-1’s therapeutic promise, particularly guiding its application within the food industry. This comprehensive approach aims to ensure that future clinical trials are well informed and strategically poised to maximize the potential of this promising probiotic strain.

## 5. Conclusions

Integrating *B. fragilis*, particularly strain SNBF-1, into food products aims to support gut health, digestion, and infection prevention. However, its dual nature as a beneficial probiotic and potential pathogen necessitates careful management. Our study highlights SNBF-1’s safety and benefit for metabolic health concerning in vitro sugar and lipid metabolism. These insights contribute to the larger goal of leveraging gut microbiota for health maintenance and disease prevention, underlining the importance of meticulous research in developing safe, effective probiotic treatments.

## Figures and Tables

**Figure 1 foods-13-00735-f001:**
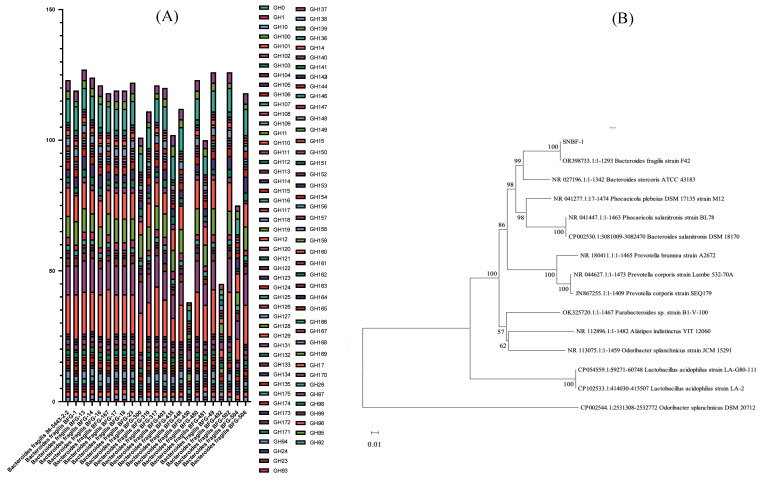
Enzymatic profiles and phylogenetic relationships of *B. fragilis* strains. (**A**) Stacked histogram of carbohydrate active enzymes (CAZy) of 23 *B. fragilis*. (**B**) Phylogenetic relation of SNBF-1 with other relevant strains.

**Figure 2 foods-13-00735-f002:**
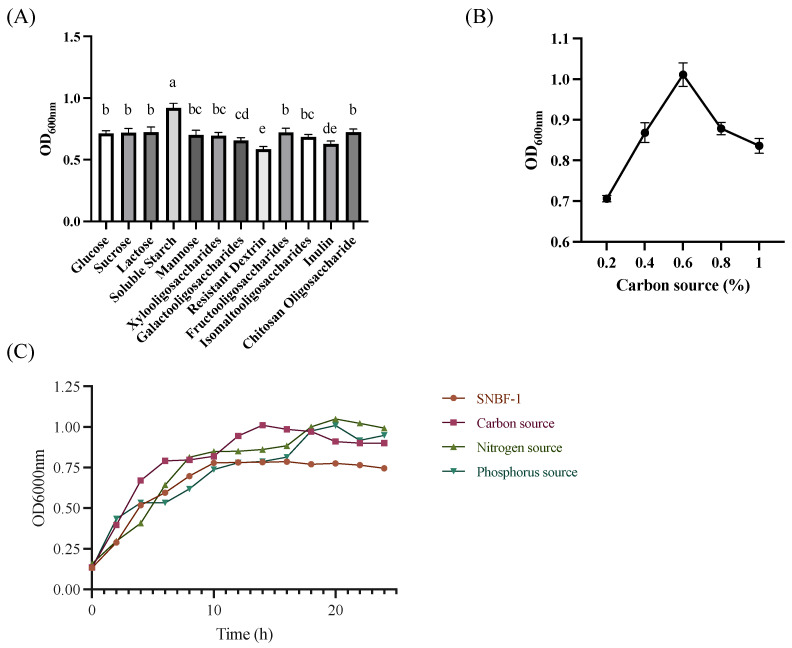
The effect of different culture conditions on the growth curve of SNBF-1. (**A**) The impact of different carbon sources on bacterial density. (**B**) The exclusive selection of starch and its impact on SNBF-1, The vertical lines reflect data variability and represents standard deviation. (**C**) Based on BHI as the base medium, we enhanced it with soluble starch as the carbon source, soy peptone as the nitrogen source, and dipotassium hydrogen phosphate as the phosphate source. Different letters (a–e) indicate significant difference (*p* < 0.05), *n* = 3.

**Figure 3 foods-13-00735-f003:**
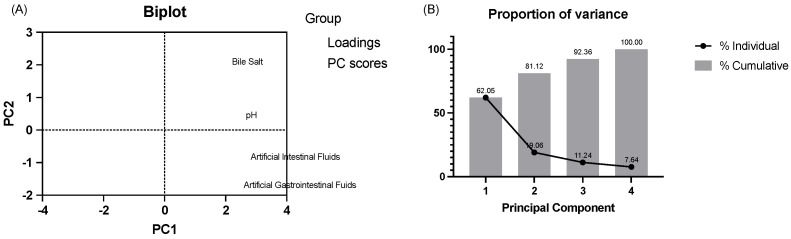
The survival rate of *B. fragilis* strains in bile salt, HCl, AGF, and AIF. (**A**) Biplot of PCA loadings and scores for bacterial survival in different solutions. (**B**) Variance explained by principal components in bacterial survival data analysis.

**Figure 4 foods-13-00735-f004:**
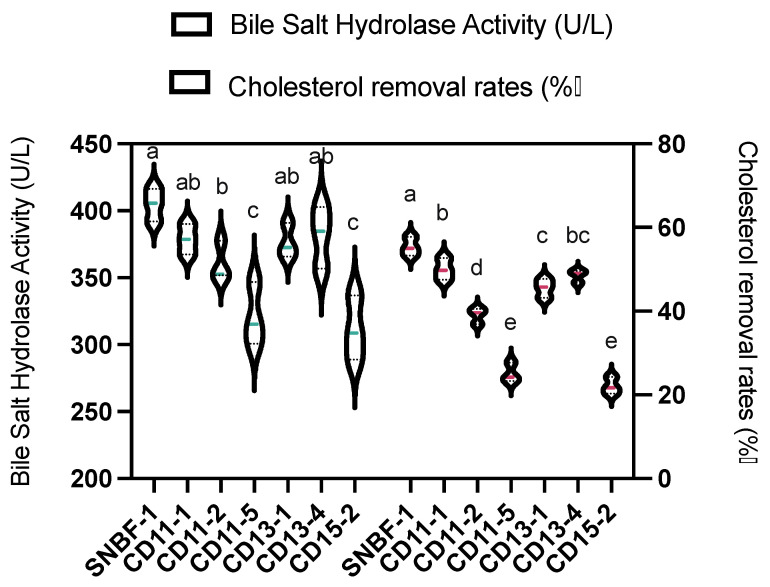
Determination of cholesterol-lowering ability of *B. fragilis*. The image depicts the in vitro capability of *B. fragilis* strains to diminish cholesterol. The light green color represents the strains’ removal rate, while the light red color indicates bile salt hydrolase activity. The data are presented against a standard curve (y = 0.0093x + 0.1099, R² = 0.9949) at 450 nm absorbance. Different lowercase letters (a–e) indicate significant difference (*p* < 0.05), *n* = 3.

**Figure 5 foods-13-00735-f005:**
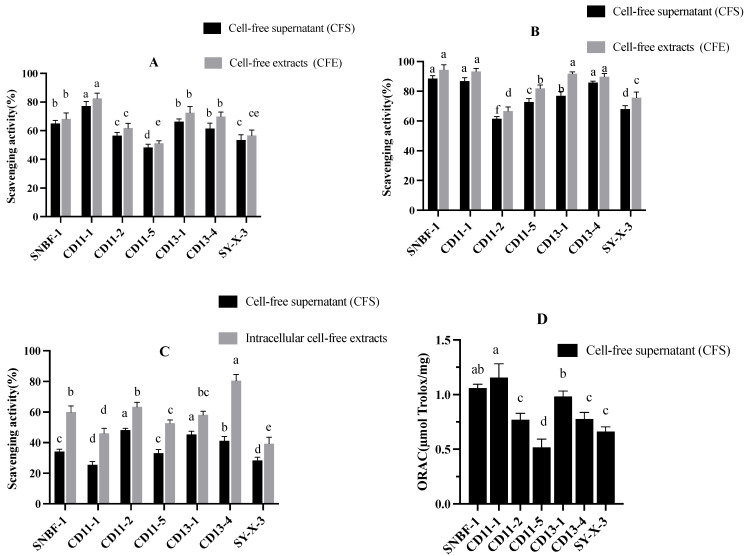
Radical scavenging and reducing capacities of different strains. (**A**) DPPH free radical scavenging capacity assay. (**B**) Hydroxyl radical scavenging capacity assay results. (**C**) ABTS free radical scavenging capacity assay results. (**D**) Reduction capacity measurement results. Different letters (a–f) indicate significant difference (*p* < 0.05), *n* = 3.

**Figure 6 foods-13-00735-f006:**
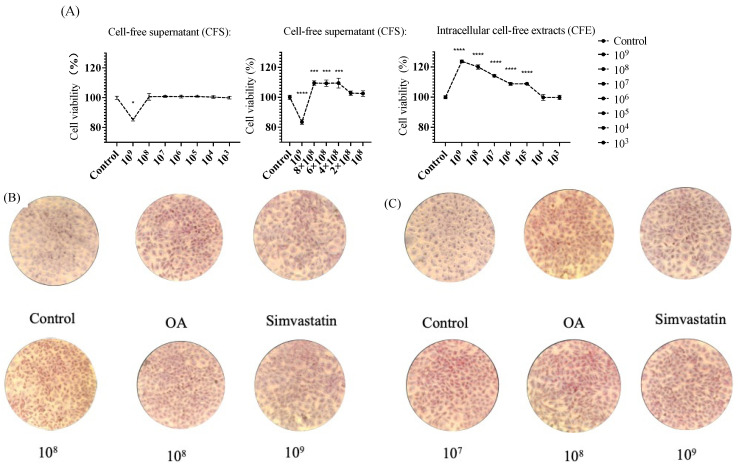
Cell viability in HepG2 cells following treatment with various concentrations of the CFS and CFE from SNBF-1. (**A**) Cell viability was assessed through a gradient dilution, spanning from 10^9^ to 10^3^, for both CFS and CFE. The concentrations tested were 10^9^, 8 × 10^8^, 6 × 10^8^, 4 × 10^8^, 2 × 10^8^, and 10^8^ cells for CFS. (**B**) Oil Red O staining was observed under the influence of CFS using an inverted microscope (**C**). Under the influence of CFE treatment, Oil Red O staining was observed using an inverted microscope. Notation (*) indicates significant difference compared with the control group (*p* < 0.05), “***” indicates a *p*-value < 0.001, and “****” denotes a *p*-value < 0.0001, both marking statistically significant differences, *n* = 3.

**Figure 7 foods-13-00735-f007:**
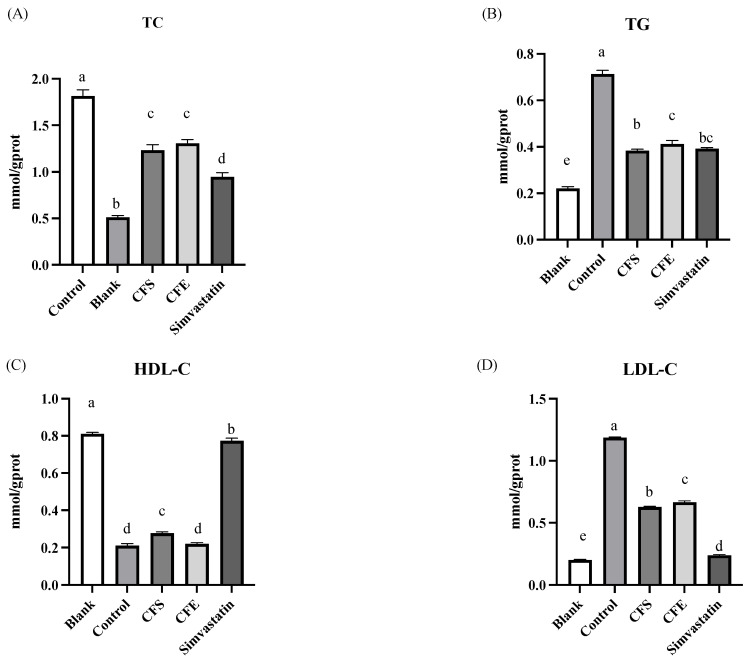
The effects of CFS and CFE from SNBF-1 on lipid accumulation. (**A**) Total cholesterol (TC), (**B**) triglycerides (TG), (**C**) high-density lipoprotein (HDL), and (**D**) low-density lipoprotein (LDL) in HepG2 cell line. Different uppercase letters (a–e) indicate significant difference (*p* < 0.05), *n* = 3.

**Figure 8 foods-13-00735-f008:**
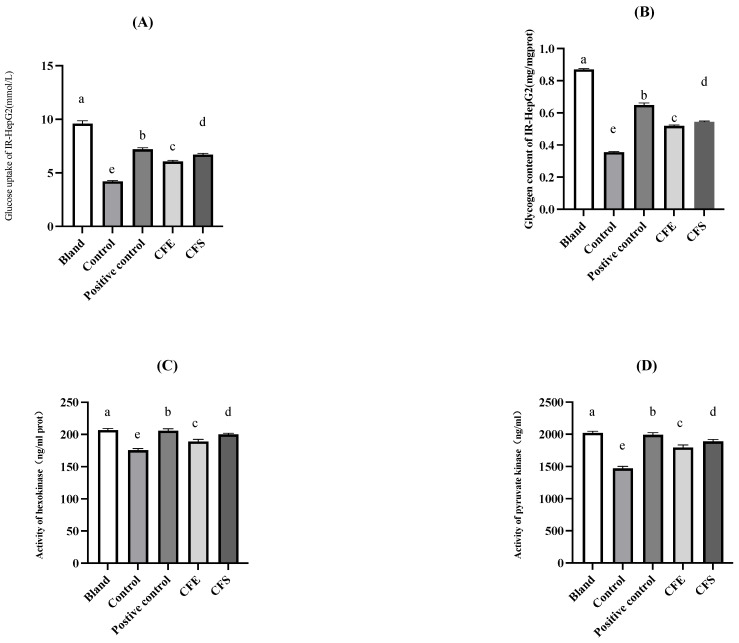
The IR-HepG2 Cell Model for antidiabetic test. (**A**) The effect of bacterial supernatant and cell-free extracts on the glucose consumption of IR-HepG2 cells is illustrated in the accompanying figure. (**B**) The results of glycogen content measurement in IR-HepG2 cells. (**C**) The activity of hexokinase (HK) in the metabolism of IR-HepG2 cells. (**D**) The measurement results of pyruvate kinase (PK) activity in the metabolism of R-HepG2 cells. Different letters (a–e) indicate significant difference (*p* < 0.05), *n* = 3.

**Table 1 foods-13-00735-t001:** Criteria for determining sensitivity of tablets to drugs.

Drug Sensitive Tablets	Drug Content (mg/Piece)	*B. fragilis* Stains						
		SNBF-1	CD11-1	CD11-2	CD11-5	CD13-1	CD13-4	SY-X-3
Ampicillin	250	R	R	R	R	R	R	R
Amoxicillin	250	I	I	R	R	R	I	R
Gentamicin	250	S	S	R	R	R	R	R
Polymyxin E	250	R	R	R	R	R	R	R
Tetracycline	100	S	I	R	R	R	I	R
Cephalosporin	100	S	S	S	S	S	S	S
Chloramphenicol	50	S	S	S	I	S	S	S
Akamycin	50	I	S	I	I	I	I	R
Ciprofloxacin	100	R	R	R	R	R	R	R
Sulfadiazine	250	R	R	R	R	R	R	R

R: resistance; I: intermediary; and S: sensitive.

**Table 2 foods-13-00735-t002:** Criteria for antibiotic resistance genes for *Bacteroides fragilis*.

Antibiotic Resistance Genes	Drug_Class	Resistance_Mechanism	Antibiotic
vanT gene in vanG cluster	Glycopeptide antibiotic	Antibiotic target alteration	Vancomycin
CepA-44	Cephalosporin	Antibiotic inactivation	-
adeF	Fluoroquinolone antibiotic; tetracycline antibiotic	Antibiotic efflux	Tetracycline
adeF	Fluoroquinolone antibiotic; tetracycline antibiotic	Antibiotic efflux	Tetracycline

**Table 3 foods-13-00735-t003:** Acid and bile salt tolerance, artificial gastrointestinal and intestinal fluids, and bile salt hydrolase activity.

Strains (%)		SNBF-1	CD11-1	CD11-2	CD11-5	CD13-1	CD13-4	SY-X-3
Acid Tolerance (pH)	2.0	90.86 ± 1.23 ^a^	94.32 ± 2.76 ^ab^	90.69 ± 0.89 ^a^	100.04 ± 3.12 ^bc^	97.57 ± 1.47 ^b^	94.32 ± 2.05 ^ab^	88.32 ± 0.92 ^a^
	3.0	92.94 ± 1.56 ^a^	103.01 ± 2.03 ^b^	91.28 ± 1.10 ^a^	108.79 ± 3.45 ^c^	98.20 ± 1.88 ^ab^	103.01 ± 2.21 ^b^	93.01 ± 1.34 ^a^
	4.0	93.46 ± 1.78 ^a^	103.28 ± 2.58 ^b^	95.00 ± 1.25 ^a^	114.78 ± 3.67 ^c^	103.32 ± 2.09 ^b^	103.28 ± 2.32 ^b^	98.28 ± 1.53 ^a^
Bile tolerance	0.1%	98.54 ± 1.34 ^Ca^	96.41 ± 2.78 ^Ba^	13.15 ± 0.43 ^Aa^	119.77 ± 2.99 ^Fa^	100.84 ± 1.15 ^Da^	129.67 ± 0.32 ^Gd^	108.14 ± 3.04 ^Ea^
	0.2%	107.14 ± 1.62 ^Cb^	104.72 ± 2.03 ^Bb^	116.94 ± 1.57 ^Eb^	122.52 ± 1.31 ^Fb^	104.19 ± 1.06 ^Ab^	112.27 ± 2.98 ^Da^	10,814.75 ± 72.07 ^Gc^
	0.3%	111.83 ± 3.33 ^Cc^	109.23 ± 1.07 ^Ac^	123.68 ± 2.81 ^Ec^	125.21 ± 2.94 ^Fc^	110.86 ± 1.34 ^Bc^	112.27 ± 1.34 ^Db^	12,667.91 ± 241.22 ^Gd^
	0.4%	124.39 ± 1.40 ^Bd^	117.94 ± 1.73 ^Ad^	140.39 ± 3.49 ^Fd^	127.77 ± 2.60 ^Cd^	129.67 ± 1.33 ^Dd^	135.36 ± 1.83 ^Ee^	13,438.85 ± 223.58 ^Ge^
	0.5%	130.27 ± 1.48 ^Ce^	125.52 ± 1.66 ^Be^	142.22 ± 3.12 ^Fe^	130.75 ± 1.94 ^De^	140.81 ± 3.22 ^Ee^	121.52 ± 2.77 ^Ac^	1132.22 ± 3.22 ^Gb^
Artificial gastrointestinal fluids		94.21 ± 1.12 ^a^	97.16 ± 0.24 ^a^	94.23 ± 0.23 ^ab^	91.74 ± 1.06 ^bc^	91.90 ± 0.74 ^bc^	93.98 ± 0.35 ^a^	94 ± 1.67 ^ab^
Artificial intestinal fluids		155.9 ± 63.70 ^a^	158.51 ± 1.14 ^a^	140.90 ± 4.17 ^b^	138.32 ± 1.53 ^bc^	148.42 ± 1.10 ^a^	94.17 ± 1.44 ^ghi^	94.17 ± 1.44 ^ghi^
Bile salt hydrolase activity		410.04 ± 12.29 ^a^	384.09 ± 11.40 ^ab^	366.02 ± 14.77 ^b^	326.31 ± 13.51 ^c^	381.85 ± 12.97 ^ab^	386.81 ± 13.17 ^ab^	316.79 ± 9.11 ^c^

Values are mean ± standard error of triplicates. Lowercase letters (a–e, g–i) indicate significant differences within the same column. Uppercase letters (A–G) indicate significant differences within the same row. The same columns are significantly different (*p* < 0.05), *n* = 3.

**Table 4 foods-13-00735-t004:** Cell surface hydrophobicity and auto-aggregation of *B. fragilis* strains.

Strains	Cell Surface Hydrophobicity (%)	Auto Aggregation (%)		
		2 h	8 h	24 h
HC-LX-1	36.41 ± 1.30 ^d^	20.67 ± 4.33	40.60 ± 0.12	41.59 ± 1.51
SNBF-1	47.23 ± 1.21 ^b^	23.44 ± 0.27	46.91 ± 2.54	52.88 ± 4.09
CD4-1	39.87 ± 1.30 ^c^	26.94 ± 4.10	49.27 ± 1.02	65.84 ± 3.19
CD11-1	45.32 ± 1.50 ^b^	31.32 ± 1.45	53.43 ± 2.76	67.80 ± 0.65
CD11-2	40.34 ± 0.63 ^c^	19.12 ± 1.07	50.54 ± 3.56	66.52 ± 6.25
CD13-4	51.48 ± 0.95 ^a^	38.33 ± 4.62	61.89 ± 1.45	77.79 ± 7.11
SY-XB-1	40.01 ± 1.29 ^c^	24.37949 ± 3.04	49.2702 ± 2.11	70.54 ± 2.88

All values are mean ± standard deviation. Values (a–d) with different superscript letters in the same column are significantly different (*p* < 0.05), *n* = 3.

## Data Availability

The original contributions presented in the study are included in the article, further inquiries can be directed to the corresponding author.

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
