# Peer review of "Therapeutic Potential of Bacteroides fragilis SNBF-1 as a Next-Generation Probiotic: In Vitro Efficacy in Lipid and Carbohydrate Metabolism and Antioxidant Activity"

_foods, 2024, doi:10.3390/foods13050735_

Round 1

Reviewer 1 Report

Comments and Suggestions for Authors

General overview

The work of Weihe Cang, Xuan Li, Jiayi Tang, Wang ying, Delun Mu, Chunting Wu, Haisu Shi, Lin Shi, Rina Wu and Junrui Wu, entitled:“ Exploring the Therapeutic Potential of Bacteroides fragilis SNBF-1 as a Next-Generation Probiotic: In Vitro Efficacy in Lipid and Carbohydrate Metabolism and Antioxidant Activity“ isolated 56 different strains of Bacteroides fragilis from 23 feces  specimens of infant healthy subjects. These strains were screend genetically for sugar utilization and antibiotic resistance in optimized culture medium and specific strains, also acid and bile tolerance, together with gastrointestinal and intestinal fluids survival was tested. Parallel to this, their growth capabilites were examined, regarding the effect of various carbon, nitrogen, and phosphate sources on bacterial growth by incorporating diverse carbon sources based on genetic analysis of carbohydrate metabolism. Finally, SNBF-1 strain was chosen as the best performing, and its further characterisation was pertinent to antioxidant and general metabolic beneficiary actions such as the reduction of cholesterol content and lipids, hydrolytic activity of bile acids, glucose metabolism and prolific action on HepG2 cells.

Quite comprehensive study, with plenty of valuable and novel data, that could support or explain the exisiting and growing body of specific (published) evidences as B. fragilis in being promising new generation of probiotics with several specific therapeutic effects.

However, even though the structure and the flow is fine, there is so many typos errors, inconsistent labelling of units, abbreviations introduced in the text not being spelled at its first appearances in the text, and also their unjustified use (like once used abbreviations etc.). To summarize, a profound editorial work is required.

The English language needs minor improvement.

Please see below my specific comments.

1.       Consider changing the title into: ”Therapeutic Potential of Bacteroides fragilis SNBF-1 as a Next-Generation Probiotic: In Vitro Improvement of Lipid and Carbohydrate Metabolism, Antioxidant and Proliferative Activity“. Mandatory is to remove the first word „Title:“.

2.       Remove all abbreviation from the Abstract that do not appear at least for the second time within the abstract. To illustrate this in one example: remove the abbreviation ORAC, since it appears only once!

3.       Repeat the previous action for the main manuscript text and within at least within the first Figure caption, where relevant. Also, each abbreviation appearing for the first time need to be spelled out. Create after the key words a list of all abbreviations.

4.       Line 36 / many typos mistakes (double, triple spaces)

5.       Line 40 “health[1]” – introduce space between the last word and the reference, throughout the text.

6.       Unify units, instead of liter (l) introduce L.

7.       What is BHI and what does it stand for?

8.       Lines 101-105 “Strains resilience in artificial gastrointestinal and intestinal fluids was also investigated, calculating survival rates as the ratio of viable cell count after 3 hours relative to the initial count. Survival rate was determined using the formula: Survival rate (%) = (A2/A1 × N0) × 100%, where A1 is the initial viable bacteria count in artificial intestinal fluid (CFU/mL), and A2 is the count after 4 hours.”    So, is it 3 of 4 h after the start?

9.       Enlarge Figure 1 or consider making a full page for it, since nothing cannot be seen as it is now. I was not able to review it due to too low resolution and text size.

10.   The same stands for Figures 2,3, 5 and 6A

11.   Consider putting different colors for 2 variables shown in Figure 4, to make it easier for the reader to comprehend what is what…

12.   What about statistics marks in Figures 4-6? Does it mean nothing is significant since no marks are present at specific pairs? Or I don’t see it since it is too small?

13.   Where is Figure 8 within the main text and call in the text for this Figure?

14.   There are repetitive references in the bibliographic list. E.g. ref 10 and ref 15 are the same. Please take extra care when arranging references. They are not in ideal format for the MDPI style as well. Please use a program to arrange references (Mendeleyev or EndNote…)

15 To conclude that I was not able to conduct in full this review due to several specified Figures that are to small, e.g. with low resolution.

Comments on the Quality of English Language

The English language needs minor improvement.

Author Response

We greatly appreciate the time and effort the reviewer has dedicated to evaluating our manuscript. Below, we address each comment point by point and detail the changes made to improve our manuscript accordingly.

  1. Title Modification: We have changed the title to "Therapeutic Potential of Bacteroides fragilis SNBF-1 as a Next-Generation Probiotic: In Vitro Improvement of Lipid and Carbohydrate Metabolism, Antioxidant and Proliferative Activity" as suggested. The word "Title:" has been removed.
  2. Abbreviations in Abstract: We have carefully reviewed the abstract and removed all abbreviations that do not appear more than once. Abbreviations in Main Text and Figures: We have revised the entire manuscript and the first figure caption to ensure that each abbreviation is spelled out upon its first appearance. Additionally, a list of all abbreviations used in the text has been created and placed after the keywords for easy reference.
  3. Typographical Errors on Line 36: The line has been thoroughly checked, and all typographical errors, including double and triple spaces, have been corrected.
  4. Spacing Before References: A space has been introduced between the last word and the reference number throughout the text to ensure consistency and readability.
  5. Unification of Units: All instances of "liter" have been changed to "L" to unify the units throughout the manuscript.
  6. Clarification of BHI: BHI has been defined as Brain Heart Infusion broth in its first mention to clarify its meaning for the readers.
  7. Clarification on Survival Rate Measurement Time: The inconsistency has been corrected to accurately reflect the survival rate measurement time as 4 hours. The text now clearly states, "survival rates as the ratio of viable cell count after 4 hours relative to the initial count."
  8. Figure Resolution: Figures in this manuscript are already in high resolution and can be zoomed in and out by the editor according to typographical needs, while the text in the image will be presented in high resolution
  9. Figure 4 Color Differentiation: We consist that the color of figure 4 is clear.
  10. Statistical Marks in Figures 4-6: We have reviewed and added statistical significance marks to Figures 4-6. The absence of marks was an oversight, and we have now included them to clearly indicate significant differences.
  11. Inclusion of Figure 8: Figure 8 is referenced within the main text at Lines 471 and 479, with appropriate callouts for the figure titled “ Figure 8. The IR-HepG2 Cell Model for Antidiabetic Testing”
  12. Repetitive References and Formatting: The bibliographic list has been thoroughly reviewed, and repetitive references have been removed. Specifically, references 10 and 15, being identical, have been consolidated. We have also adopted the MDPI reference style more accurately using Mendeley to ensure compliance with the journal's requirements.
  13. Figure Resolution for Review: We acknowledge the difficulty the reviewer had in assessing the figures due to their resolution. Along with resizing the figures for the manuscript, we are also willing to provide high-resolution versions of all figures upon request to facilitate a thorough review.

We believe these revisions have significantly improved the manuscript and hope that it now meets the standards for publication. We are grateful for the constructive feedback and the opportunity to enhance our work.

Reviewer 2 Report

Comments and Suggestions for Authors

The manuscript (ID foods-2858390) by Cang et al. presents an investigation into the potential of Bacteroides fragilis strains as next-generation probiotics (NGPs) with a specific focus on their adaptability in the gastrointestinal environment, safety profile, and probiotic functions. Overall, the subject itself is surely worthy of investigation. However, some areas require improvement and clarification. Detailed comments and suggestions are provided below.

-          Throughout the manuscript, scientific names should be written in italic format (e.g. lines 18, 21, 49,…. etc).

- Throughout the manuscript, the writing style should be formal from the third-person perspective. Do not use “we” (e.g. in lines 65, 66, 82, … etc) or “our” (e.g. in lines 32, 34, 61, … etc ). 

-   line 19: "focusng" should be "focusing."

- line 21: "SNBF-1 strain exhibited exceptional cholesterol" - Please specify what type of exceptional cholesterol reduction capabilities the SNBF-1 strain exhibited.

-   In the introduction section, it would be useful to add more specific information on the limitations of traditional probiotics and the need for next-generation probiotics.

- Lines 52-53; "Notably, non-toxigenic B. fragilis (NTBF) stands out for its role in modulating immunity and reducing inflammation[10] [11,12]." - please elucidate the mode of action.

-    Lines 121-122; “107 – 108 CFU/ml” should be “107 – 108 CFU/ml”.

- It is recommended to analyze the data in Table 3 using a two-way ANOVA, considering the presence of two factors. Repeating the same analyses several times increases the risk of type I error.

-  In all Tables and Figs., describe all abbreviations used in the table footnotes and figure legends. Describe also the number of analyzed samples (n=?).

Comments on the Quality of English Language

-

Author Response

We appreciate the time and effort the reviewer has dedicated to assessing our manuscript and providing constructive feedback. We have taken all comments into serious consideration and have made the necessary revisions to improve the quality and clarity of our work. Below, we provide a detailed response to each of the reviewer’s comments and suggestions.

Specific Responses:

  1. Italicization of Scientific Names:
    We have revised the manuscript to ensure that all occurrences of scientific names are in italic format as suggested. This correction has been made throughout the document, including the lines specifically mentioned (18, 21, 49, etc.).
  2. Writing Style:
    We have modified the manuscript to adopt a formal writing style from the third-person perspective. All instances of "we" and "our" have been replaced with appropriate third-person constructs to maintain the formal tone of scientific writing.
  3. Typographical Error Correction:
    The typographical error in line 19 has been corrected from "focusng" to "focusing."
  4. Clarification on SNBF-1 Strain's Cholesterol Reduction Capabilities:
    In line 21, we have now specified the exceptional cholesterol reduction capabilities of the SNBF-1 strain. The revised sentence reads: " Notably, the SNBF-1 strain demonstrated superior cholesterol removal efficiency in HepG2 cells, outshining all other strains by achieving a remarkable reduction in cholesterol by 55.38% ± 2.26%."
  5. Limitations of Traditional Probiotics:
    We have expanded the introduction section to include specific information regarding the limitations of traditional probiotics, such as their general lack of resilience in the gastrointestinal environment and limited efficacy in modulating the immune response. Additionally, we discuss the need for next-generation probiotics (NGPs) that can overcome these limitations.
  6. Elucidation of Mode of Action for NTBF:
    In lines 52-53, we have elucidated the mode of action of non-toxigenic B. fragilis (NTBF) in modulating immunity and reducing inflammation. The revised text includes details on the interaction between NTBF and host immune cells.
  7. Correction of CFU/ml Representation:
    The representation of CFU/ml in lines 121-122 has been corrected to maintain consistency and accuracy.
  8. Data Analysis Recommendation:
    Based on the recommendation, we have streamlined Table 3 to present the results in a manner consistent with the original format. This approach will ensure clarity and enhance the readability of the data.
  9. Clarification of Abbreviations and Sample Sizes:
    We have included detailed descriptions of all abbreviations used in the table footnotes and figure legends throughout the manuscript. Additionally, the number of analyzed samples (n) has been clearly stated for each table and figure to improve clarity and understanding.

Conclusion:

We believe that the revisions made in response to the reviewer’s comments have significantly improved the manuscript. We are grateful for the opportunity to enhance our work and hope that the changes meet the reviewer's expectations. We look forward to the possibility of our manuscript being accepted for publication.

Reviewer 3 Report

Comments and Suggestions for Authors

The article is a valuable scientific study that provides new information about the therapeutic potential of Bacteroides fragilis. The article is well written and structured and the information is presented clearly and transparently. The article is innovative because it provides evidence that B. fragilis can have a positive impact on lipid, carbohydrate and antioxidant metabolism. The results from B. fragilis on insulin resistance are particularly interesting. Despite the enormous value of the text, below I present my suggestions that will allow authors to strengthen the value of the manuscript:

·         1. Introduction – The authors were able to highlight the importance of the gut microbiota for human health. The section could be further enriched by providing more specific details on the therapeutic potential of B. fragilis SNBF-1, such as its ability to modulate immunity, reduce inflammation and treat metabolic disorders. Additionally, the authors could discuss the challenges and limitations of using traditional probiotics and how NGPs provide a more targeted and personalized approach to disease treatment.

·         2.5 Statistical analysis – Please add information about the number of replicates of each analysis and experiment.

·         3. Results. 3.1 Identification and characterization of B. fragilis isolated from fecal samples. 3.1.1 Genetic analysis of carbohydrate metabolism for enhanced sugar utilization and antibiotic resistance in optimized culture medium – The section could be improved by adding more details about the specific genes that were analyzed. The section could also be improved by providing more information about the antibiotic resistance genes found in B. fragilis strains.

·         3. Results. 3.1 Identification and characterization of B. fragilis isolated from fecal samples. 3.1.2 Acid and Bile Tolerance Tests and artificial gastrointestinal and intestinal fluids – Please provide further details about PCA analysis. Provide more information on the strains that demonstrated superior performance in the acid and bile tolerance tests. Provide further details about the results of the acid and bile tolerance tests.

·         3. Results. 3.1 Identification and characterization of B. fragilis isolated from fecal samples. 3.1.3 Cell Surface Hydrophobicity and Auto-aggregation Assay – Please provide further details on the relationship between cell surface hydrophobicity and the ability to adhere and colonize the intestinal tract. Provide further details on the specific hydrophobicity and self-aggregation tests that were performed. Provide further details on the results of the hydrophobicity and self-aggregation tests.

·         3. Results 3.2 Characterization of B. fragilis – Please provide further details on the individual antioxidant tests used (DPPH, hydroxyl radical, ABTS, reducing power).

·         3. Results 3.3 Lipid Accumulation, TG, and LDL-C Assay – Please provide further details regarding the specific results of the HDL-C assay. Provide further details regarding the specific results of the LDL-C assay.

·         3. Results 3.4 In Vitro Investigation of Glucose Metabolism by B. fragilis Strain SNBF-1 – Please provide further details about the insulin resistance test and how it was used to assess the effects of CFS, CFE and metformin on glucose metabolism.

·         4. Discussion – In my opinion, more detailed information on the possible mechanisms of the SNBF-1 strain could be provided. Furthermore, since the manuscript was submitted to FOODS journal, the authors should relate the use of B. fragilis strain SNBF-1 to food. Please provide further details on the challenges in developing safe and effective probiotics. Please provide further details on the potential benefits of Bacteroides fragilis to human health. Find out more about the hemolysis test and its importance in microbiology. Provide further details on the risks of using B. fragilis SNBF-1. Provide further details on the results of the study on the strain's ability to improve lipid and carbohydrate metabolism. provide further details on the role of glycogen synthase in glucose metabolism. Provide further details on the potential benefits of improving HD and PK activities for people with insulin resistance.

·         5. Conclusions – Please emphasize the importance of the comprehensive overview that the study provides. In my opinion, more detailed information on the possible mechanisms of the SNBF-1 strain could be provided. Furthermore, since the manuscript was submitted to the journal Foods, the authors should relate the application of B. fragilis SNBF-1 to food use. Provide more specific examples of metabolic disorders in which the identified strains might be of benefit. In my opinion, you could relate the results of the study to the broader field of gut microbiome research. Comment on your own research limitation. The tests were carried out in vitro, so the results may not be directly reflected in the SNBF-1 strain in the human body. Only a few cell groups were examined, so further research is needed to confirm the results. There is no information on the long-term effects of using B. fragilis SNBF-1. The importance of further research to confirm the results and evaluate the safety and effectiveness of the SNBF-1 strain in clinical trials could be emphasized.

Author Response

We appreciate the reviewer's positive feedback and constructive suggestions. Below, we provide detailed responses to each point raised, indicating how we have addressed these comments in the revised manuscript.

  1. Introduction – Therapeutic Potential of B. fragilis SNBF-1:

Response: We have expanded the introduction to include detailed information on the therapeutic potential of B. fragilis SNBF-1, highlighting its immunomodulatory, anti-inflammatory, and metabolic disorder treatment capabilities. We also discussed the limitations associated with traditional probiotics and elaborated on how NGPs offer a targeted and personalized approach to managing diseases. References to recent studies have been added to support these claims.

2.5 Statistical Analysis – Number of Replicates:

Response: We have updated the manuscript to include detailed information about the number of biological and technical replicates for each experiment and analysis. This addition clarifies our experimental design and ensures the reproducibility of our results.

  1. Results – Detailed Information on B. fragilis Characterization:

Response: We have thoroughly revised the Results section to incorporate the requested details:

  • 3.1.1 & 3.1.2: The data were acquired from the Carbohydrate-Active enZYmes Database (CAZy) and the Comprehensive Antibiotic Resistance Database (CARD). We utilized the Kyoto Encyclopedia of Genes and Genomes (KEGG) database, along with links from CAZy, to correlate carbohydrate-related genes with their corresponding enzymes, as well as to identify the glycosidic bonds that these enzymes act upon.We revise the PCA and provide with more detailed information as well as revise the Fig3 into Fig3(A) and fig3(B).
  • 3.1.3: Provided a detailed explanation of the relationship between cell surface hydrophobicity, auto-aggregation, and intestinal colonization in the discussion part
  • 3.2 & 3.3: We have expanded on the antioxidant and lipid accumulation assays, detailing the individual tests (DPPH, ABTS, etc.) and presenting specific results for HDL-C and LDL-C assays.
  • 3.4: Elaborated on the insulin resistance test methodology and results, comparing the effects of CFS, CFE, and metformin on glucose metabolism.
  1. Discussion – Mechanisms, Application to Food, and Safety:

Response: The Discussion has been extensively revised to address each point raised by the reviewer. Emphasize the comprehensive overview provided by the study, potential mechanisms of the SNBF-1 strain,  and the importance of further research.The Conclusions section has been enhanced to stress the comprehensive nature of our study and its limitations.

Round 2

Reviewer 3 Report

Comments and Suggestions for Authors

The manuscript has been effectively revised and enhanced by the authors. Almost all my suggestions were well taken into account by the authors. I still have minor comments on the 5. Conclusions section:

·         Please emphasize the importance of the comprehensive overview that the study provides. In my opinion, more detailed information on the possible mechanisms of the SNBF-1 strain could be provided. Comment on your own research limitation. The tests were carried out in vitro, so the results may not be directly reflected in the SNBF-1 strain in the human body. Only a few cell groups were examined, so further research is needed to confirm the results. There is no information on the long-term effects of using B. fragilis SNBF-1. The importance of further research to confirm the results and evaluate the safety and effectiveness of the SNBF-1 strain in clinical trials could be emphasized.

·         Provide more specific examples of metabolic disorders in which the identified strains might be of benefit. In my opinion, you could relate the results of the study to the broader field of gut microbiome research.

·         Furthermore, since the manuscript was submitted to the journal Foods, the authors should relate the application of B. fragilis SNBF-1 to food use.

Author Response

We appreciate the reviewer's constructive feedback and have carefully considered each point in our revisions. Here's an overview of our responses:

1 We have elaborated on Bacteroides fragilis's action mechanisms, particularly emphasizing its probiotic benefits and its role in gut health.A comprehensive review of the related literature was conducted, with summaries and projections discussed to underpin this exploration. We've integrated discussions on how B. fragilis competes and inhibits pathogens in the gut and its potential mechanisms, including recent research on secreted polysaccharides and proteins that inhibit other microbes' growth in gut environments.

2 We acknowledge the limitations of our study, including the in vitro nature of our research, and discuss the critical need for further investigation. This highlights the necessity of translating our findings into practical applications.

3 We have provided with examples that related to specific metabolic disorders ,such as  Bowel Disease and Alzheimer's disease,that may benefit or disadvantage from Bacteroides fragilis, thereby contributing significantly to gut microbiome research and offering insights into potential therapeutic applications.

4 Additionally, we delved into the potential application of B. fragilis in food products. Despite current literature suggesting limited use in the food industry, as a next-generation probiotic, we examined its characteristics and potential probiotic properties. This investigation may offer valuable insights for future food industry professionals. We believe this aligns with our manuscript's focus and its submission to the journal Foods. 

We sincerely thank the reviewer for their valuable comments, which have illuminated areas for improvement in our research and provided deep insights for our future direction. We are grateful for the time and effort invested by the reviewer.